# Population risk factors for severe disease and mortality in COVID-19: A global systematic review and meta-analysis

Adam Booth[1]⊚, Angus Bruno Reed[1]⊚, Sonia Ponzo[1], Arrash Yassaee[1], Mert Aral[1], David Plans [1,2]*, Alain Labrique[3], Diwakar Mohan[3]

1 Huma Therapeutics Limited, London, United Kingdom, 2 INDEX Group, Department of Science, Innovation, Technology, and Entrepreneurship, University of Exeter, Exeter, United Kingdom, 3 Johns Hopkins Bloomberg School of Public Health, Baltimore, MD, United States of America

⊚ These authors contributed equally to this work.
* david.plans@huma.com

**Data Availability Statement:** All relevant data are within the manuscript and its Supporting Information files.

## Abstract

### Aim

COVID-19 clinical presentation is heterogeneous, ranging from asymptomatic to severe cases. While there are a number of early publications relating to risk factors for COVID-19 infection, low sample size and heterogeneity in study design impacted consolidation of early findings. There is a pressing need to identify the factors which predispose patients to severe cases of COVID-19. For rapid and widespread risk stratification, these factors should be easily obtainable, inexpensive, and avoid invasive clinical procedures. The aim of our study is to fill this knowledge gap by systematically mapping all the available evidence on the association of various clinical, demographic, and lifestyle variables with the risk of specific adverse outcomes in patients with COVID-19.

### Methods

The systematic review was conducted using standardized methodology, searching two electronic databases (PubMed and SCOPUS) for relevant literature published between 1st January 2020 and 9th July 2020. Included studies reported characteristics of patients with COVID-19 while reporting outcomes relating to disease severity. In the case of sufficient comparable data, meta-analyses were conducted to estimate risk of each variable.

### Results

Seventy-six studies were identified, with a total of 17,860,001 patients across 14 countries. The studies were highly heterogeneous in terms of the sample under study, outcomes, and risk measures reported. A large number of risk factors were presented for COVID-19. Commonly reported variables for adverse outcome from COVID-19 comprised patient characteristics, including age >75 (OR: 2.65, 95% CI: 1.81–3.90), male sex (OR: 2.05, 95% CI: 1.39–3.04) and severe obesity (OR: 2.57, 95% CI: 1.31–5.05). Active cancer (OR: 1.46, 95% CI: 1.04–2.04) was associated with increased risk of severe outcome. A number of common

**Funding:** This research was funded by Huma Therapeutics Ltd. The funders had no role in study design, data collection and analysis, decision to publish, or preparation of the manuscript.

**Competing interests:** A.B, A.B.R., S.P., D.P., A.Y., M.A., are employees of Huma Therapeutics Ltd. D. M & AL declare that they have no conflict of interests to report. This does not alter our adherence to PLOS ONE policies on sharing data and materials.

symptoms and vital measures (respiratory rate and SpO2) also suggested elevated risk profiles.

## Conclusions

Based on the findings of this study, a range of easily assessed parameters are valuable to predict elevated risk of severe illness and mortality as a result of COVID-19, including patient characteristics and detailed comorbidities, alongside the novel inclusion of real-time symptoms and vital measurements.

## Introduction

SARS-CoV-2, first reported to the WHO on 31 December 2019, has subsequently exponentially spread with cases now officially reported in 215 countries and territories [1]. Following infection, individuals may develop COVID-19, an influenza-like illness targeting, primarily, the respiratory system. The clinical pathophysiology of COVID-19 is still the subject of ongoing research. It is clear, however, that clinical presentation is heterogeneous, ranging from asymptomatic to severe disease. Common clinical features include major symptoms such as fever, cough, dyspnoea [2], and minor symptoms such as altered sense of smell and taste [3, 4], gastrointestinal symptoms [5], and cutaneous manifestations [6]. Evidence suggests most patients move through two phases: (a) viral replication over several days with relatively mild symptoms; (b) adaptive immune response stage, which may cause sudden clinical deterioration [7]. Severe symptoms are thought to be the consequence of the SARS-CoV-2 virus invading type II alveolar epithelial cells, causing the release of cytokines and inflammatory markers. This 'cytokine storm' attracts neutrophils and T cells, which in turn cause significant lung injury and inflammation, eventually leading to acute respiratory distress syndrome [8]. There are a number of different classifications of COVID-19, with recent attempts to sub-divide intensive care patients into different clinical phenotypes [9]. Guidelines for the classification of COVID-19 disease severity in adults were first reported in February 2020 and have since been widely adopted internationally [10]. Reported complications and long-term sequelae in survivors are varied and include neurologic, hematologic, musculoskeletal, cardiovascular, and GI-related issues [11]. While most patients recover quickly, a growing number are suffering from so-called 'long COVID', a multisystem, post-viral condition with symptoms including fatigue, anxiety, low mood, cognitive problems, and atypical chest pain, stretching over a period of weeks or months without recovery [12]. In addition, mental health conditions (e.g. PTSD, depression, and anxiety) are also known to result from extended ICU admission [13].

COVID-19 has posed unprecedented care and logistic challenges, with resource-intense care settings such as critical care having to increase capacity by up to 300% [14]. This has significant downstream effects on wider healthcare capacity, including the delivery of elective surgical care and mental health services [15]. For example, DATA-CAN estimates that the impact of reducing access to cancer screening, triage, and treatment will result in a further 7,165–17,910 excess deaths amongst the UK population within one year [16]. It is for this reason that many national strategies have focused, from the outset, on preventing health systems becoming overloaded by clinical demand [17].

COVID-19 has posed unprecedented care and logistic challenges, with resource-intense care settings such as critical care having to increase capacity by up to 300% [9]. This has significant downstream effects on wider healthcare capacity, including the delivery of elective surgical care and mental health services [10]. For example, DATA-CAN estimates that the impact

of reducing access to cancer screening, triage, and treatment will result in a further 7,165–17,910 excess deaths amongst the UK population within one year [11]. It is for this reason that many national strategies have focused, from the outset, on preventing health systems becoming overloaded by clinical demand [12].

The ability to predict the likelihood of severe health outcomes in patients affected by COVID-19 has the potential to inform decision-making at the individual, provider, and government level. At the patient level, accurate prognostication could facilitate evidence-based decisions around shielding. At a provider level, predictors of severity, if coupled with epidemiological models, could enable accurate scenario planning and inform resource allocation decisions. At a governmental level, population-wide risk assessments could help inform the targeted use of non-pharmacological interventions, potentially minimising the economic and population health impact of wide-sweeping social distancing measures. Furthermore, with news that national governments have begun procuring COVID-19 vaccines, an evidence-based risk stratification tool could help policymakers decide which segments of the population to prioritise in national vaccination programmes [18].

Although serologic biomarkers are useful in grading the severity of a COVID-19 case upon admission to the hospital, patients are often experiencing severe disease by the time they present clinically. The ability to stratify cases earlier in the disease process (based on demographics and lifestyle factors) could prove invaluable to initiating earlier referrals and possibly improving patient outcomes. To allow rapid and widespread risk stratification, these factors should be easily obtainable, inexpensive, and avoid invasive clinical procedures. These factors should also help shape decision-making at an individual, provider, and system level. To this end, we included symptom information in our analysis on the grounds that individuals isolating at home with COVID-19, along with their clinical team, can be informed about their risk of deterioration as and when new symptoms develop. Retrospective cohort data suggest that many patients present to hospital more than seven days after onset of symptoms, potentially offering providers some, albeit short, notice to prioritise resources if necessary [19]. In contrast, blood tests are only likely to be of value in stratifying disease severity amongst those patients already severe enough to require hospitalisation. Blood test data would, therefore, provide limited use for individuals' behaviour modification or remote monitoring, and is unlikely to help providers anticipate increased clinical demand.

However, there are challenges in creating such a prognostic tool based on individual or small numbers of studies. While the volume of academic reporting on clinical features of COVID-19 has been unprecedented, low sample size and heterogeneity in study design impacted the consolidation of early findings. Early reports on clinical features were limited to Wuhan, China [20], and the lack of geographical, cultural, and ethnic diversity has restricted the generalisability of findings. As such, the aim of our study is to fill this knowledge gap by systematically mapping all the available evidence on the association of various clinical, demographic, and lifestyle variables with the risk of specific adverse outcomes in patients with COVID-19.

## Methods

This protocol is in line with the recommendations outlined in the Preferred Reporting Items for Systematic Reviews and Meta-Analyses (PRISMA) statement.

### Eligibility criteria

Peer-reviewed observational studies published between 1st January 2020 and 9th July 2020 in the English language were included. Only papers reporting original data on adult (>16 years

old) patients with laboratory-confirmed SARS-CoV-2 were selected. The minimum sample size for inclusion was 100 patients. Narrative reviews, case reports, papers only reporting laboratory or imaging data, and papers not reporting original data were not included. Studies including homogeneous populations with exclusion criteria (e.g. female patients pregnant at the time the study was conducted) were also excluded.

## Information sources and search strategy

A systematic review using PubMed and SCOPUS was conducted. Additionally, a thorough hand search of the literature and review of the references of included papers in the systematic review was carried out to minimize the likelihood that the used search terms did not identify all relevant papers. The following search terms were included: ncov* OR coronavirus OR "SARS-CoV-2" OR "covid-19" OR covid, AND ventilator OR ICU OR "intensive care" OR mortality OR prognosis OR ARDS OR severity OR prognosis OR hospitalis* OR hospitaliz* OR "respiratory failure" OR intubation OR ventilation OR admission* OR admitted OR "critical care" OR "critical cases", AND clinical OR symptom* OR characteristic* OR comorbidit* OR co morbidit* OR risk OR predict* and "PUBYEAR > 2019". Comprehensive search terms can be found in supplementary material (S1 Table).

## Study selection

Two authors (A.B.R. and A.B.) independently reviewed titles and abstracts to ascertain that all included articles were in line with the inclusion criteria (Fig 1). Studies with missing, unclear, duplicated, or incomplete data were excluded from the review. Observational studies including original data on at least 100 adult patients with laboratory-confirmed SARS-CoV-2, whether hospitalised or in outpatient settings, were included in the meta-analysis.

## Data collection process and data items

The following information was extracted from each selected article: author, publication year, article title, location of study, SARS-CoV-2 case identification, study type (e.g. primary research, review, etc), peer-review status, quality assessment, and total sample size. Extracted data included sample demographics (age, sex, ethnicity), obesity/BMI status, smoking status, blood type, any existing comorbidities, symptoms, basic clinical variables (e.g. heart rate, respiration rate, and oxygen saturation), and their clinical outcomes of severe (severe case definition, admission to ICU, invasive mechanical ventilation (IMV), and death) versus non-severe comparator event (e.g. no ICU admission, survival/recovery). Data extraction was carried out using software specifically developed for systematic review (Covidence, Veritas Health Innovation, Melbourne, Australia).

## Assessment of methodological quality and risk of bias

An adapted version of the Newcastle-Ottawa Scale [21] was used during full-text screening to assess the methodological quality of each article. Two authors reviewed the quality of included studies (A.B.R. and A.B.), with conflicts resolved in consensus. Studies were judged on three criteria: selection of participants; comparability of groups; and ascertainment of the exposure and outcome of interest.

## Statistical approach

Reported measures relating to patient characteristics, comorbidities, symptoms, and vital signs were extracted from included articles. We analysed similar risk metrics for each outcome and

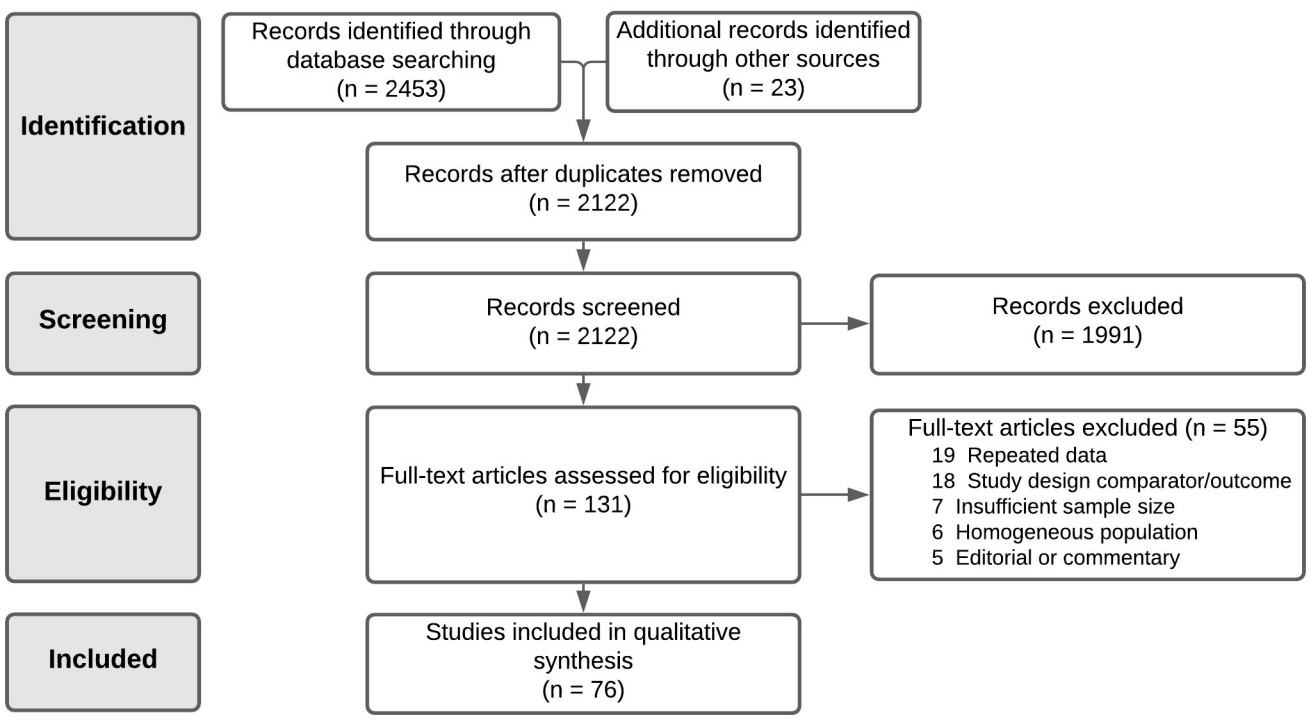

**Fig 1. PRISMA diagram.**

pooled extracted values. Where possible, a meta-analysis was carried out to assess the strength of association between reported risk factors and two outcomes: severe and mortality. Severe outcome was defined as the clinical definition of severe, ICU admission, or IMV, while excluding hospitalisation. If a study reported multiple outcomes, then the clinical definition of severe [22, 23] was taken to avoid duplication of data. Meta-analysis regression of reported multivariate Odd Ratios (ORs) were pooled with estimated effect size calculated using a random-effects model.

To accommodate for heterogeneity across the studies, we estimated risk weighting for each reported variable across two endpoints: severe COVID-19 (comprising severe case definition, ICU admission, and IMV) and mortality from COVID-19. If at least two studies reported ORs (multivariate or univariate) for the same clinical variable, pooled weighted estimates were calculated on the basis of sample size and standard error. If only a single study reported the finding, a point estimate from that study was listed. Data were analysed using the R statistical software [24]. The meta-analysis and plots were created using the R package *meta* [25].

## Results

The comprehensive search of databases and cross-referencing hand search identified 2122 articles meeting the search criteria, following removal of duplicates. During screening of title and abstract, 1991 articles were excluded. Consequently, 131 articles were selected for full-text review. Of these, 76 articles were deemed to meet the inclusion/exclusion criteria. Articles were excluded for the following primary reasons: repeated data (n = 19); wrong design/outcome of interest (n = 18); insufficient sample size (n = 7); homogenous population (n = 6); and

editorial or commentary (n = 5) (full reasoning is noted in Fig 1). A summary of all included studies' characteristics and quality assessment is given in Table 1. Inter-rater reliability of article inclusion was substantial (κ = 0.74).

Research was pooled from 14 geographies with China the most commonly reported (n = 43) [19, 26–67], followed by USA (n = 15) [68–82], Italy (n = 4) [83–86], and UK (N = 3) [87–89]. The remaining papers were from Mexico [90, 91], South Korea [92, 93], Turkey [94], Brazil [95], Denmark [96], France [97], Israel [98], Iran [99], and Poland [100]. A total of 17,860,001 subjects are described in the included studies; however, after excluding two large national cohort studies which involved non-COVID-19 subjects [88, 89], the final sample included data on 153,115 reported individuals with COVID-19.

Reported outcomes across studies varied and were categorised into five grouped endpoints: severe, hospitalisation, ICU admission, IMV, composite endpoint (considered as ICU, IMV, or mortality), and mortality.

The literature reported a wide variety of variables that may provide insight to estimate risk of adverse outcomes in COVID-19. These variables were grouped into four categories: patient characteristics, comorbidities, presenting symptoms, and vital signs (Table 1). Univariate ORs were reported, or calculated where sufficient data was presented, in 65 articles. Multivariate ORs were reported in 45 studies, while 17 reported Hazard Ratios and two reported Risk Ratios.

Meta-analysis regression was carried out to investigate the pooled risk estimates of selected factors for severe outcome. Analysed patient characteristics included age >75, male sex, and severe obesity (BMI>40) (Fig 2). Age >75 years old was an important factor contributing to severe outcomes in COVID-19 (OR: 2.65, 95% CI: 1.81–3.90, $I^2$ = 51%). Males had higher risk compared to females (OR: 2.05, 95% CI: 1.39–3.04, $I^2$ = 75%). Severely obese individuals were at higher risk compared to non-severely obese individuals (OR: 2.57, 95% CI: 1.31–5.05, $I^2$ = 39%). When considering mortality as the outcome, the risk associated with age >75 is elevated further (OR: 5.57, 95% CI: 3.10–10.00, $I^2$ = 28%) (S1 Fig).

The risk associated with pre-existing conditions including hypertension, diabetes, active cancer, and chronic kidney disease (CKD) was also investigated using meta-analysis (Fig 3). Active cancer (OR: 1.46, 95% CI: 1.04–2.04, $I^2$ = 0%) was associated with increased risk of severe outcome. Diabetes (OR: 1.99, 95% CI: 0.92–4.29, $I^2$ = 43%), Hypertension (OR: 1.33, 95% CI: 0.99–1.80, $I^2$ = 63%), and CKD (OR: 1.27, 95% CI: 0.70–2.29, $I^2$ = 88%) showed no significant elevated risk. Forest plots showing meta-analysis regression for the relative risk of mortality conferred by hypertension, diabetes, and active cancer are reported in S2 Fig. To highlight the heterogeneity of reported outcomes in included studies, all reported risk estimates for male sex, diabetes and hypertension as presented as an example in the supplementary material (S3–S5 Figs respectively).

Due to the heterogeneity of studies and insufficient comparable data, it was not possible to conduct meta-regression on all reported variables, including symptoms and vitals measurements. As such, pooled weighted estimates were extracted where possible (Table 2). Further patient characteristics such as blood type A and smoking history shows trends towards elevated risk for severe outcome (OR: 1.45, OR: 1.42, 95% CI: 1.41, 1.43, respectively). A number of symptoms suggested elevated risk of severe outcome including myalgia (OR: 4.82. 95% CI: 4.63–5.01), sputum production (OR: 11.40), dyspnoea (OR: 8.68, 95% CI: 8.25–9.11), nausea (OR: 15.55), and chills (OR: 6.32). Fever showed low estimated risk for both severity and mortality (OR: 1.06, OR: 0.69 respectively). There was insufficient comparable data to estimate risk for cough as an independent factor, however, pooling univariate analysis also found low estimated risk for both severity and mortality (OR: 1.01, OR: 1.08 respectively). There was limited evidence on loss of smell as a risk factor for severe outcomes [93].

**Table 1.** Summary of studies.

| Study | Publication date | Country | Study design | Sample size (n) | Severe | Non-severe comparator | Patient characteristics | Comorbidities | Symptoms | Vital signs | Study Quality (mNOS) |
|---|---|---|---|---|---|---|---|---|---|---|---|
| Argenziano 2020 [68] | 29/05/2020 | United States | Retrospective single-centre, case series | 1,000 | ICU | ER, hospital (non-ICU) | Male, Age, BMI, Smoking, Ethnicity | Any comorbidity, Hypertension, CVD, Diabetes, Cerebrovascular Disease, Chronic Liver Disease, Asthma, COPD, Sleep Apnoea, Active Cancer, Interstitial Lung Disease, CKD, Transplant history, Rheumatic Disease, Chronic Lung Disease, Viral Hepatitis, HIV | Fever, Myalgia, Cough, Sputum, Dyspnoea, Nausea, Diarrhoea, Pharyngalgia, Headache, Chills, Rhinorrhoea | | 6 |
| Baqui 2020 [95] | 02/07/2020 | Brazil | Retrospective multi-centre, cross sectional study | 11,321 | Mortality | Survived/Recovered | Male, Age, BMI, Ethnicity | CVD, Diabetes, Chronic Liver Disease, Chronic Lung Disease, Asthma, Immunosuppression, CKD, Neurological Disease | | | 9 |
| Bello-Chavolla 2020 [90] | 01/07/2020 | Mexico | Retrospective multi-centre, cross sectional study | 51,633 | Mortality | Survived/Recovered | Age, BMI | Diabetes, Chronic Lung Disease, Immunosuppression, CKD | Haemoptysis | | 10 |
| Cao 2020 [26] | 13/03/2020 | China | Retrospective single-centre, case series | 102 | Mortality | Survived/Recovered | Male | Any comorbidity, Hypertension, CVD, Cerebrovascular Disease, Chronic Liver Disease, Chronic Lung Disease, Active Cancer, CKD | Fever, Fatigue, Myalgia, Cough, Diarrhoea | | 6 |
| Chen 2020 [27] | 19/03/2020 | China | Retrospective single-centre, case series | 249 | ICU | Non-ICU | Male, Age | Any comorbidity | | | 10 |
| Chen 2020 [28] | 26/03/2020 | China | Retrospective single-centre, case series | 274 | Mortality | Survived/Recovered | Male, Age, Smoking | Any comorbidity, Hypertension, CVD, Diabetes, Chronic Lung Disease, Active Cancer, Immunosuppression, CKD, Chronic GI, Viral Hepatitis | Fever, Fatigue, Myalgia, Cough, Sputum, Dyspnoea, Chest pain, Nausea, Pharyngalgia, Headache, Dizziness, GI, Anorexia | Respiratory Rate, Heart Rate, Oxygen saturation % | 6 |

(*Continued*)

**Table 1.** (Continued)

| Study | Publication date | Country | Study design | Sample size (n) | Severe | Non-severe comparator | Patient characteristics | Comorbidities | Symptoms | Vital signs | Study Quality (mNOS) |
|---|---|---|---|---|---|---|---|---|---|---|---|
| Chen 2020 [29] | 16/06/2020 | China | Retrospective multi-centre, case series | 1,859 | Mortality | Survived/Recovered | Age, Smoking | | Fever | | 9 |
| Cummings 2020 [69] | 19/05/2020 | United States | Retrospective multi-centre, case series | 257 | Mortality | Survived/Recovered | Male, Age | Hypertension, CVD, Diabetes, Chronic Lung Disease | | | 9 |
| D'Silva 2020 [70] | 26/05/2020 | United States | Retrospective, single-centre comparative cohort study | 156 | 1. Hospitalisation<br>2. Composite endpoint: Mechanical ventilation/ intensive care admission<br>3. Mortality | 1. Non-hospitalisation<br>2. Non-mechanical ventilation/ intensive care admission<br>3. Survival | | Rheumatic Disease | | | 9 |
| Dai 2020 [30] | 28/04/2020 | China | Retrospective, multi-centre comparative cohort study | 641 | 1. Severe symptoms<br>2. Intensive care admission<br>3. Invasive Mechanical Intervention<br>4. Mortality | 1. Mild symptoms<br>2. Non-intensive care admission<br>3. Non-invasive Mechanical Intervention<br>4. Survival | | Active Cancer | | | 6 |
| Deng 2020 [31] | 25/02/2020 | China | Retrospective multi-centre, case series | 225 | Mortality | Survived/Recovered | Male | Any comorbidity, Hypertension, CVD, Diabetes, Chronic Lung Disease | Fever, Fatigue, Cough, Sputum, Dyspnoea | | 7 |
| Docherty 2020 [87] | 22/05/2020 | UK | Prospective multi-centre cohort study | 20,133 | Mortality | Discharged | Male, Age, BMI | CVD, Diabetes, Chronic Liver Disease, COPD, Active Cancer, CKD, Dementia, Neurological Disease | Diarrhoea | | 10 |
| Du 2020 [32] | 08/04/2020 | China | Retrospective single-centre, case series | 179 | Mortality | Survived/Recovered | Male, Age | Hypertension, CVD, Diabetes, Chronic GI | Fatigue, Myalgia, Cough, Sputum Dyspnoea, Headache, GI | | 8 |
| Ellinghaus 2020 [86] | 17/06/2020 | Italy and Spain | Retrospective multi-centre, genome-wide association study | 3,815 | Respiratory failure | No respiratory failure | Blood Type | | | | 10 |

*(Continued)*

**Table 1.** (Continued)

| Study | Publication date | Country | Study design | Sample size (n) | Severe | Non-severe comparator | Patient characteristics | Comorbidities | Symptoms | Vital signs | Study Quality (mNOS) |
|---|---|---|---|---|---|---|---|---|---|---|---|
| Feng 2020 [33] | 01/06/2020 | China | Retrospective multi-centre, case series | 476 | Severe & critical disease (5th ed. COVID-19 guidelines NHC) | Moderate disease (5th ed. COVID-19 guidelines NHC) | Male, Age, Smoking, Alcohol Intake | Any comorbidity, Hypertension, CVD, Diabetes, Cerebrovascular Disease, COPD, Immunosuppression, CKD, Others | Fever, Myalgia, Cough, Sputum, Dyspnoea, Chest pain, Pharyngalgia, GI, Haemoptysis, Chills | | 7 |
| Göker 2020 [94] | 23/06/2020 | Turkey | Retrospective single-centre, case series | 186 | Composite endpoint: Intubation, ICU or Mortality | Undefined | Blood Type | | | | 7 |
| Giacomelli 2020 [83] | 22/05/2020 | Italy | Prospective single-centre, case series | 233 | Mortality | Survived/ Recovered | Male, Age, BMI, Smoking | Any comorbidity | Fever, Cough, Dyspnoea, Nausea | Haemoglobin Levels | 10 |
| Grasselli 2020 [84] | 28/04/2020 | Italy | Retrospective multi-centre, case series | 1,591 | Mortality | Discharged or still in ICU | Age | Hypertension | | | 6 |
| Guan 2020 [34] | 14/05/2020 | China | Retrospective multi-centre, case series | 1,590 | Composite endpoint: ICU, Intubation or Mortality | Non-ICU or survivor | | Any comorbidity, Hypertension, CVD, Diabetes, Cerebrovascular Disease, COPD, Active Cancer, Immunodeficiency, CKD, Others | | | 10 |
| Gupta 2020 [71] | 06/08/2020 | United States | Retrospective multi-centre, case series | 2,215 | Mortality within 28 day of ICU admission | Survival within 28 day of ICU admission | Male, Age, BMI, Smoking, Ethnicity | Any comorbidity, Hypertension, CVD, Diabetes, Chronic Lung Disease, Asthma, COPD, Active Cancer, Immunodeficiency, CKD | Fever, Fatigue, Cough, Sputum, Nausea | | 9 |
| Hajifathalian 2020 [72] | 05/08/2020 | United States | Retrospective multi-centre, case series | 770 | Composite endpoint: ICU or Mortality | Non-ICU or survivor | Age, BMI, Ethnicity | | | | 10 |
| Hou 2020 [92] | 23/06/2020 | South Korea | Retrospective single-centre, case series | 211 | Progression to severe stage COVID-19 | Asymptomatic or mildly symptomatic patients who were discharged | Male, Age | Hypertension, Diabetes | Fever, Myalgia, Cough, Sputum, Dyspnoea, Chest pain, Diarrhoea, Pharyngalgia, Headache, Chills, Rhinorrhoea | | 10 |
| Huang 2020 [35] | 08/05/2020 | China | Retrospective multi-centre, case series | 202 | Severe disease (5th ed. COVID-19 guidelines NHC) | Non-severe disease (5th ed. COVID-19 guidelines NHC) | Male, Age, BMI, Smoking | Any comorbidity, Hypertension, Chronic Heart Disease, Diabetes | Fever, Fatigue, Cough, Dyspnoea, Pharyngalgia | | 9 |

(Continued)

Table 1. (Continued)

| Study | Publication date | Country | Study design | Sample size (n) | Severe | Non-severe comparator | Patient characteristics | Comorbidities | Symptoms | Vital signs | Study Quality (mNOS) |
|---|---|---|---|---|---|---|---|---|---|---|---|
| Huang 2020 [36] | 01/06/2020 | China | Retrospective multi-centre, case series | 310 | 1. Severe (5th ed. COVID-19 guidelines NHC) 2. Mortality | 1. Non-severe 2. Survival | Male, Age | Hypertension | Nausea | | 10 |
| Imam 2020 [73] | 04/06/2020 | United States | Retrospective multi-centre, case series | 1,305 | Mortality | Survived/Recovered | Male, Age, Smoking | Hypertension, CVD, Diabetes, Cerebrovascular Disease, Chronic Liver Disease, Asthma, COPD, Sleep Apnoea, Active Cancer, Immunosuppression, CKD, Dementia | | | 10 |
| Israelsen 2020 [96] | 15/05/2020 | Denmark | Retrospective single-centre, case series | 175 | ICU | General ward treatment | | | | | 6 |
| Itelman 2020 [98] | 01/05/2020 | Israel | Retrospective single-centre, case series | 162 | Severe—defined as requiring intensive help for proper oxygenation (high-flow oxygen delivery device or artificial ventilation, either non-invasive or invasive) | Mild or Moderate disease (flu-like without clinical and imaging signs of pneumonia; pneumonia and hypoxemia) | Male | Hypertension, Chronic Heart Disease, Diabetes | | | 4 |
| Jin 2020 [37] | 01/06/2020 | China | Retrospective multi-centre, case series | 651 | Severe/Critical disease (6th ed. COVID-19 guidelines NHC) | Mild/Moderate disease (6th ed. COVID-19 guidelines NHC) | | | Sputum, GI | | 9 |
| Kalligeros 2020 [74] | 02/06/2020 | United States | Retrospective multi-centre, case series | 103 | 1. ICU admission within the first 10 days 2. IMV during the first 10 days | 1. No ICU admission within the first 10 days 2. No IMV during the first 10 days | Male, Age, BMI, Smoking, Ethnicity | Hypertension, Chronic Heart Disease, Diabetes, Chronic Lung Disease | | | 10 |

(Continued)

**Table 1.** (Continued)

| Study | Publication date | Country | Study design | Sample size (n) | Severe | Non-severe comparator | Patient characteristics | Comorbidities | Symptoms | Vital signs | Study Quality (mNOS) |
|---|---|---|---|---|---|---|---|---|---|---|---|
| Kammar-Garcia 2020 [91] | 25/05/2020 | Mexico | Retrospective multi-centre, case series | 13,842 | 1. Mortality / 2. Composite endpoint: Hospitalization, pneumonia, intubation, and ICU admission | 1. Survival / 2. Outpatient | BMI | Hypertension, CVD, Diabetes, Asthma, COPD, Immunosuppression, CKD | | | 9 |
| Kim 2020 [75] | 16/07/2020 | United States | Retrospective multi-centre, case series | 2,490 | 1. ICU / 2. Mortality | Hospitalisation without event | Male, Age, BMI, Smoking, Ethnicity | Hypertension, CVD, Diabetes, Chronic Lung Disease, Immunosuppression, CKD, Neurological Disease, Rheumatic Disease | | | 10 |
| Lassale 2020 [88] | 01/06/2020 | UK | Retrospective multi-centre, cohort study | 428,494 | Hospitalisation | Non-hospitalised | Male, Age, BMI, Smoking, Ethnicity, Alcohol Intake | Hypertension, CVD, Chronic Lung Disease | | | 10 |
| Latz 2020 [76] | 12/07/2020 | United States | Retrospective multi-centre, case series | 1,289 | Composite endpoint: intubation and death | Hospitalisation without event | Blood Type | | | | 10 |
| Lee 2020 [93] | 06/05/2020 | Korea | Retrospective multi-centre, case series | 3,191 | Severe and critical disease (Daegu Severity Score for COVID-19) | Mild & moderate disease | | | Loss of smell/taste | | 6 |
| Li 2020 [101] | 08/04/2020 | China | Retrospective multi-centre, case series | 132 | Mortality | Survived/Recovered | Male, Age | | | | 10 |
| Li 2020 [39] | 29/05/2020 | China | Retrospective single-centre, case series | 453 | Mortality | Survived/Recovered | | Diabetes | | | 9 |
| Li 2020 [40] | 11/06/2020 | China | Retrospective multi-centre, case series | 1,449 | Mortality | Survived/Recovered | Male, Age, Smoking | | Fatigue, Myalgia, Cough, Sputum, Dyspnoea, Nausea, Diarrhoea, Headache, Chills | Haemoglobin Levels | 6 |
| Liang 2020 [41] | 12/05/2020 | China | Retrospective multi-centre, case series | 1,590 | Composite endpoint: ICU, ventilation, or death | Hospitalisation without event | Male, Age, Smoking | Any comorbidity, Hypertension, CVD, Diabetes, Cerebrovascular Disease, COPD, Active Cancer, CKD | Fever, Fatigue, Myalgia, Cough, Dyspnoea, Pharyngalgia, Headache, Haemoptysis, Chills, Unconsciousness | | 10 |

*(Continued)*

Table 1. (Continued)

| Study | Publication date | Country | Study design | Sample size (n) | Severe | Non-severe comparator | Patient characteristics | Comorbidities | Symptoms | Vital signs | Study Quality (mNOS) |
|---|---|---|---|---|---|---|---|---|---|---|---|
| Liu 2020 [43] | 14/04/2020 | China | Retrospective single-centre, case series | 140 | Severe (7th ed. COVID-19 guidelines NHC) | Mild disease | Male, Age | Hypertension, CVD | Fever, Fatigue, Myalgia, Cough, Dyspnoea, Chest pain, Anorexia | | 6 |
| Liu 2020 [42] | 27/04/2020 | China | Retrospective single-centre, case series | 134 | Severe (7th ed. COVID-19 guidelines NHC & American Thoracic Society) | Non-severe disease | Male | Hypertension, Diabetes | Fever, Fatigue, Cough, Sputum, Anorexia | | 8 |
| Masetti 2020 [85] | 14/06/2020 | Italy | Retrospective single-centre, case series | 229 | Mortality | Discharged survivors | Male, Age | Any comorbidity, Hypertension, Chronic Heart Disease, Diabetes, COPD, Active Cancer, CKD | | | 9 |
| Nowak 2020 [100] | 18/05/2020 | Poland | Retrospective single-centre, case series | 169 | Mortality | Survived/Recovered | Male, Age | Hypertension, CVD, Diabetes, COPD, Active Cancer, CKD, Others | Fever, Fatigue, Cough, Dyspnoea, Nausea, Diarrhoea | | 8 |
| Okoh 2020 [77] | 10/06/2020 | United States | Retrospective single-centre, case series | 251 | Mortality | Survived/Recovered | Male, Age, Ethnicity | Hypertension, CVD, Diabetes, Cerebrovascular Disease, COPD, Active Cancer, CKD | Fever | Respiratory Rate, Heart Rate, Haemoglobin Levels | 9 |
| Palaiodimos 2020 [78] | 15/05/2020 | United States | Retrospective single-centre, case series | 200 | 1. Increasing Oxygen 2. Intubation 3. Mortality | ICU admission without event | Male, Age, BMI, Smoking, Ethnicity, Alcohol Intake | Hypertension, CVD, Diabetes, Cerebrovascular Disease, Asthma, COPD, Sleep Apnoea, Active Cancer, Immunosuppression, CKD | | | 9 |
| Pei 2020 [44] | 29/05/2020 | China | Retrospective single-centre, case series | 333 | Severe/Critical (7th ed. COVID-19 guidelines NHC) | Moderate (7th ed. COVID-19 guidelines NHC) | Male | Hypertension, Diabetes | Fever, Cough, Dyspnoea, Diarrhoea | | 6 |
| Petrilli 2020 [79] | 01/05/2020 | United States | Retrospective single-centre, case series | 5,279 | 1. Hospitalisation 2. Composite endpoint: intensive care unit, mechanical ventilation, discharge to hospice, or death | Non-hospitalised; alive | Male, Age, BMI, Smoking, Ethnicity | Hypertension, CVD, Diabetes, Asthma, COPD, Active Cancer, CKD, Hyperlipidaemia | Fever, Fatigue | Oxygen saturation % | 10 |

(Continued)

Table 1. (Continued)

| Study | Publication date | Country | Study design | Sample size (n) | Severe | Non-severe comparator | Patient characteristics | Comorbidities | Symptoms | Vital signs | Study Quality (mNOS) |
|---|---|---|---|---|---|---|---|---|---|---|---|
| Price-Haywood 2020 [80] | 25/06/2020 | United States | Retrospective multi-centre, case series | 3,481 | 1. Hospitalisation<br>2. Composite endpoint: intensive care unit, mechanical ventilation, discharge to hospice, or death | Non-hospitalised; alive | Male, Age, BMI, Ethnicity | | | Respiratory Rate | 10 |
| Qin 2020 [45] | 29/05/2020 | China | Retrospective multi-centre, case series | 1,875 | 1. Severe<br>2. Mortality | Non-hospitalised; alive | | Cerebrovascular Disease | | | 8 |
| Ramlall 2020 [81] | 03/08/2020 | United States | Retrospective multi-centre, case series | 6,393 | 1. Intubation<br>2. Mortality | Hospitalisation without event | Age, BMI, Smoking | Hypertension, CVD, Diabetes, Coagulation disorder, Macular Degeneration | Cough | | 10 |
| Ren 2020 [46] | 11/05/2020 | China | Retrospective single-centre, case series | 151 | Severe (6th ed. COVID-19 guidelines NHC) | Mild (6th ed. COVID-19 guidelines NHC) | Male | Hypertension, CVD, Diabetes | Fever, Fatigue, Cough, Sputum, Dyspnoea, Nausea, Diarrhoea, Anorexia | | 10 |
| Ruan 2020 [47] | 03/03/2020 | China | Retrospective single-centre, case series | 150 | Mortality | Survived/Recovered | Male | Hypertension, CVD, Diabetes, Cerebrovascular Disease | Fever, Fatigue, Myalgia, Cough, Sputum, Dyspnoea | | 5 |
| Shahriarirad 2020 [99] | 18/06/2020 | Iran | Retrospective single-centre, case series | 113 | 1. Severe (American Thoracic Society)<br>2. Mortality | Non-severe; alive | Male | Hypertension, CVD, Diabetes | Fever, Fatigue, Myalgia, Cough, Sputum, Dyspnoea, Chest pain, Nausea, Diarrhoea, Headache, Dizziness, Chills, Anorexia | Oxygen saturation % | 6 |
| Shi 2020 [48] | 18/03/2020 | China | Retrospective single-centre, case series | 487 | Severe (undefined) | Mild (undefined) | Male, Age | Hypertension | | | 9 |
| Shi 2020 [49] | 28/04/2020 | China | Retrospective multi-centre, case series | 306 | Mortality | Survived/Recovered | Male | Hypertension, CVD | Fever, Fatigue, Cough, Dyspnoea, Anorexia | | 5 |
| Simonnet 2020 [97] | 09/04/2020 | France | Retrospective single-centre, cohort study | 124 | Ventilation | ICU with no mechanical ventilation | Male, Age, BMI | Hypertension, Diabetes | | | 9 |

(Continued)

**Table 1.** (Continued)

| Study | Publication date | Country | Study design | Sample size (n) | Severe | Non-severe comparator | Patient characteristics | Comorbidities | Symptoms | Vital signs | Study Quality (mNOS) |
|---|---|---|---|---|---|---|---|---|---|---|---|
| Suleyman 2020 [82] | 16/06/2020 | United States | Retrospective single-centre, case series | 463 | 1. Hospitalisation 2. ICU 3. Mechanical ventilation | Hospitalisation without event | Male, Age, BMI, Smoking, Ethnicity | Hypertension, CVD, Diabetes, Asthma, COPD, Sleep Apnoea, Active Cancer, CKD | Fever, Myalgia, Cough, Dyspnoea, Nausea, Diarrhoea, Headache, Loss of smell/taste, Anorexia | Respiratory Rate | 10 |
| Wang 2020 [52] | 20/02/2020 | China | Retrospective single-centre, case series | 138 | ICU | Non-ICU | Male | Any comorbidity, Hypertension, CVD, Diabetes, Cerebrovascular Disease, COPD, Active Cancer, CKD | Fatigue, Myalgia, Cough, Sputum, Dyspnoea, Nausea, Diarrhoea, Pharyngalgia, Headache, Dizziness, Anorexia | | 6 |
| Wang 2020 [51] | 30/03/2020 | China | Retrospective single-centre, case series | 339 | Mortality | Survived/ Recovered (at 4 weeks) | Male | Hypertension, CVD, Diabetes, Cerebrovascular Disease, COPD | Fever, Fatigue, Cough, Sputum, Dyspnoea, Diarrhoea, Anorexia | | 7 |
| Wang 2020 [50] | 08/04/2020 | China | Retrospective single-centre, case series | 344 | Mortality | Survived/ Recovered | Male, Age | Hypertension, CVD, Diabetes, COPD | Fever, Fatigue, Cough, Sputum, Dyspnoea, Diarrhoea, Anorexia | | 10 |
| Wang 2020 [53] | 11/04/2020 | China | Retrospective single-centre, case series | 125 | Critical (5th ed. COVID-19 guidelines NHC) | Non-critical | Male | Any comorbidity | | | 9 |
| Wang 2020 [54] | 30/04/2020 | China | Retrospective single-centre, case series | 107 | Mortality | Survived/ Recovered | Male, Age | Hypertension, CVD | | | 7 |
| Williamson 2020 [89] | 08/07/2020 | UK | Retrospective multi-centre, cohort study | 17,278,392 | Mortality | Survived/ Recovered | Male, Age, BMI, Smoking, Ethnicity | Hypertension, Chronic Heart Disease, Diabetes, Cerebrovascular Disease, Chronic Liver Disease, Chronic Lung Disease, Asthma, Active Cancer, Immunosuppression, CKD, Dementia, Neurological Disease, Transplant history, Rheumatic Disease, Chronic GI | | | 8 |

*(Continued)*

Table 1. (Continued)

| Study | Publication date | Country | Study design | Sample size (n) | Severe | Non-severe comparator | Patient characteristics | Comorbidities | Symptoms | Vital signs | Study Quality (mNOS) |
|---|---|---|---|---|---|---|---|---|---|---|---|
| Wu 2020 [55] | 19/05/2020 | China | Retrospective single-centre, case series | 1,048 | Composite endpoint: ICU, mechanical ventilation, or death | | | COPD | | | 9 |
| Xie 2020 [56] | 13/04/2020 | China | Retrospective single-centre, case series | 140 | Mortality | Survived/Recovered | | Any comorbidity, Hypertension | Dyspnoea | Oxygen saturation % | 8 |
| Yan 2020 [57] | 06/04/2020 | China | Retrospective single-centre, case series | 193 | Mortality | Survived/Recovered | Male | Hypertension, Diabetes | | | 6 |
| Yang 2020 [58] | 25/05/2020 | China | Retrospective single-centre, case series | 200 | ICU | Non-ICU | Male, Age, Smoking | Any comorbidity, Hypertension, Chronic Heart Disease, Diabetes, Chronic Lung Disease, Active Cancer, CKD | Fever, Fatigue, Myalgia, Cough, Dyspnoea, Nausea, Diarrhoea, Pharyngalgia, Headache, Chills | | 7 |
| Yao 2020 [59] | 24/04/2020 | China | Retrospective single-centre, case series | 108 | Severe (American Thoracic Society) | Non-severe (American Thoracic Society) | Male, Age, Smoking | Any comorbidity, Hypertension, CVD, Diabetes, Chronic Liver Disease | Fever, Fatigue, Cough, Sputum, Dyspnoea, Diarrhoea | | 10 |
| Ye 2020 [60] | 13/06/2020 | China | Retrospective multi-centre, case series | 856 | 1. Severe (6th ed. COVID-19 guidelines NHC) 2. ICU 3. Mortality | Mild; hospitalised non-event | | Any comorbidity, Hypertension, Chronic Heart Disease, Diabetes, Active Cancer, CKD, Viral Hepatitis, Others | | | 10 |
| Yu 2020 [61] | 27/04/2020 | China | Retrospective multi-centre, case series | 421 | Composite severity outcome (ICU, ARDS, or shock) | No composite endpoint | Male, Age | Hypertension, Chronic Heart Disease, Diabetes | Fever, Cough, Sputum | | 9 |
| Zhang 2020 [64] | 15/03/2020 | China | Retrospective multi-centre, case series | 645 | Severe / Critical (5th ed. COVID-19 guidelines NHC) | Mild to moderate disease (5th ed. COVID-19 guidelines NHC) | Male | Any comorbidity | Fever, Fatigue, Myalgia, Cough, Sputum, Dyspnoea, Nausea, Diarrhoea, Pharyngalgia | | 10 |
| Zhang 2020 [62] | 05/04/2020 | China | Retrospective single-centre, case series | 221 | Severe (American Thoracic Society) | Non-severe (American Thoracic Society) | Male, Age | Any comorbidity, Hypertension, CVD, Cerebrovascular Disease, Chronic Liver Disease, COPD, Active Cancer, Immunosuppression, CKD | Fever, Fatigue, Cough, Dyspnoea, Chest pain, Diarrhoea, Pharyngalgia, Headache, Anorexia | | 8 |

(Continued)

**Table 1.** (Continued)

| Study | Publication date | Country | Study design | Sample size (n) | Severe | Non-severe comparator | Patient characteristics | Comorbidities | Symptoms | Vital signs | Study Quality (mNOS) |
|---|---|---|---|---|---|---|---|---|---|---|---|
| Zhang 2020 [63] | 15/04/2020 | China | Retrospective single-centre, case series | 663 | 1. Severe COVID-19 (National Health Commission definition (trial version 5))  2. Mortality | 1. Mild/ Moderate COVID-19 (National Health Commission definition (trial version 5))  2. Survival | Male, Age | CVD, Chronic Lung Disease, Active Cancer, Endocrine System Disease, Endocrine System Disease, Chronic GI | Fever, Fatigue, Myalgia, Cough, Sputum, Dyspnoea, Chest pain, Nausea, Diarrhoea, Headache, Dizziness | Haemoglobin Levels | 9 |
| Zhang 2020 [65] | 26/04/2020 | China | Retrospective single-centre, case series | 111 | Composite endpoint: ICU or death. | Discharge | Male | Any comorbidity, Hypertension, Diabetes | Fever, Fatigue, Myalgia, Cough, Dyspnoea, Chest pain, Diarrhoea | Respiratory Rate | 10 |
| Zheng 2020 [102] | 24/03/2020 | China | Retrospective single-centre, case series | 161 | Severe COVID-19 (National Health Commission definition (trial version 5)) | Non-severe COVID-19 (National Health Commission definition (trial version 5)) | Male | Hypertension, CVD, Diabetes, Cerebrovascular Disease, COPD | Fever, Fatigue, Myalgia, Cough, Dyspnoea, Diarrhoea, Headache | | 7 |
| Zhou 2020 [19] | 09/03/2020 | China | Retrospective single-centre, case series | 191 | Mortality | Survived/ Recovered | Male, Age, Smoking | Any comorbidity, Hypertension, CVD, Diabetes, Chronic Lung Disease, Other | Fever, Fatigue, Myalgia, Cough, Sputum, Nausea, Diarrhoea | Respiratory Rate, Haemoglobin Levels | 10 |
| Zhou 2020 [67] | 18/05/2020 | China | Retrospective single-centre, case series | 366 | Severe (American Thoracic Society) | Non-severe (American Thoracic Society) | Male, Age | Hypertension, COPD, Diabetes | Fever, Fatigue, Cough, Dyspnoea | Respiratory Rate, Heart Rate, Oxygen saturation % | 9 |

Summary of studies included in quantitative synthesis. Abbreviations: ARDS, Acute respiratory distress syndrome; BMI, Body Mass Index; COPD, Chronic Obstructive Pulmonary Disease; CKD, Chronic Kidney Disease; CVD, Cardiovascular Disease; GI, Gastrointestinal; HIV, Human Immunodeficiency Virus; ICU, Intensive Care Unit; IMV, invasive mechanical intubation; mNOS, modified Newcastle Ottawa Scale; NHC, National Health Commission

### Age >75 years old

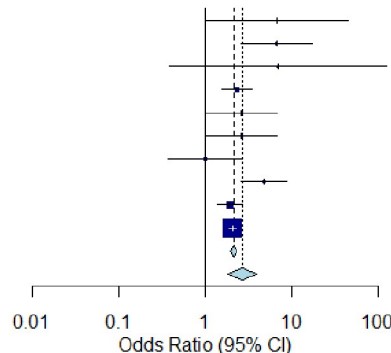

### Male

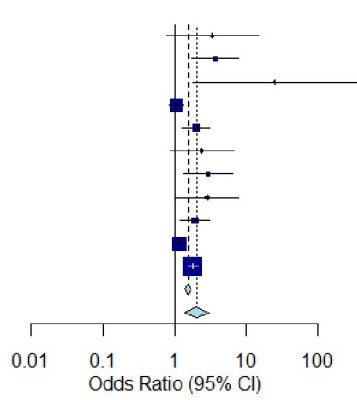

### Severe Obesity

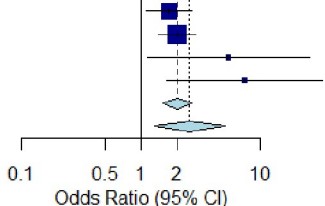

**Fig 2. Forest plot for the association of patient characteristics (age, sex, and severe obesity) with severe outcomes from COVID-19 using a random-effects model.**

Respiratory rate ≥24 breaths/min was reported as a risk in five studies [28, 49, 53, 70, 77]. However, it was not possible to combine data and provide estimates for risk due to heterogeneous outcomes and risk measures reported, with a wide range in the effect estimates (OR: 1.74, 95% CI: 0.95–3.18 vs OR: 11.60, 95% CI: 3.34–40.27). The only study carrying out multivariable analysis for respiratory rate ≥24 breaths/min found increased risk with reported OR of 2.00 95% CI:1.34–2.99 [70]. Similarly, there was insufficient data to report pooled estimates for oxygen saturation. Two studies reported multivariate analysis for mortality as outcome, showing increased risk with decreasing oxygen saturation, SpO2 88–92% (OR: 1.46, 95% CI: 1.18–1.79) and SpO2 <88% (OR: 2.00, 95% CI: 1.61–2.48) [79]. Xie et al. report that SpO2 ≤90% was strongly associated with death, independently of age and sex (hazard ratio: 47.41, 95% CI: 6.29–357.48) [56]. Univariate analysis also showed increased risk of severe outcome with SpO2 on admission to hospital <90% (OR: 3.83, 95% CI: 1.05–14.01) [99] and <93% (OR: 13.12, 95% CI: 7.11–24.24) [28].

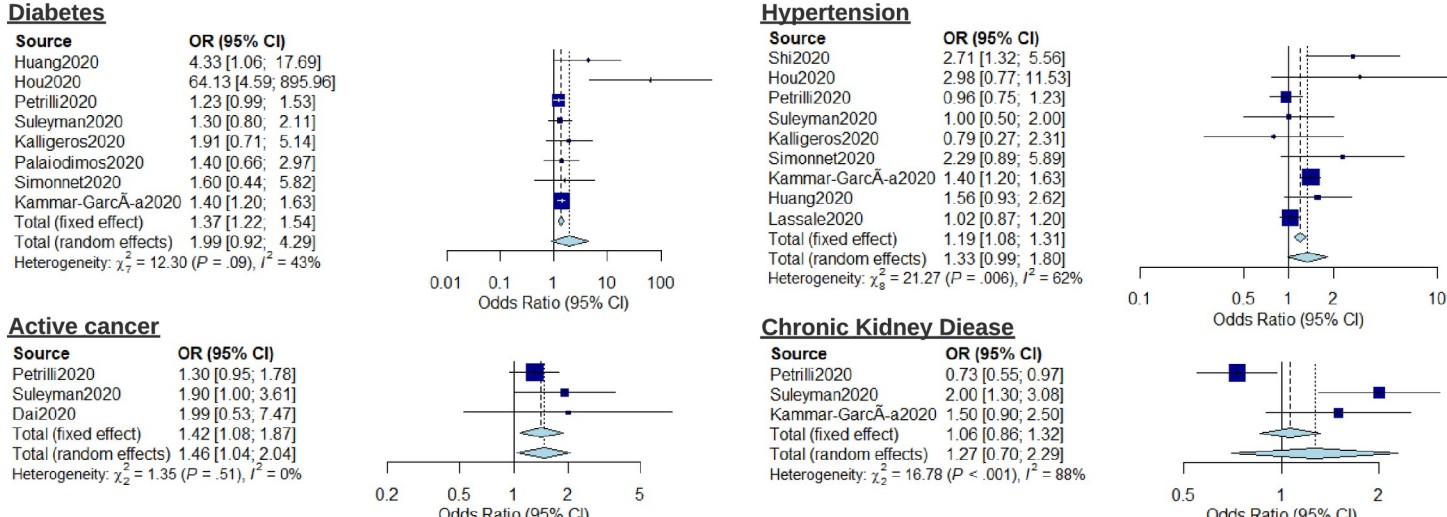

**Fig 3. Forest plot for the association of comorbidities (diabetes, hypertension, chronic kidney disease, and active cancer) with severe outcomes from COVID-19 using a random-effects model.**

## Quality assessment

Methodological structure and reporting of studies varied in quality. Quality scores were evaluated using an adapted version of the NOS [21], with an average quality score of 8.4 (SD = 1.7), ranging between 4 and 10 (scale out of 10) (Table 1). All studies reported data collection from health records. Subject inclusion in reported literature was widely reported as hospital admission with positive RT-PCR (reverse transcription polymerase chain reaction) test and, therefore, most studies show bias towards inclusion of hospitalised, thus more severe, patients. Few studies reported handling of missing data and bias reporting in findings.

## Publication bias

Given the high volume of published literature, we did not include publications in grey literature such as medRxiv and bioRxiv. As inclusion was limited to studies published only in English, language bias is likely. Due to high heterogeneity and spread of data, we estimate risk of bias based on the most commonly reported variable: male sex (Fig 4). The funnel plot showed a somewhat asymmetrical distribution, which may be explained by the small number studies, therefore high probability that deviations in funnel shape occur due to chance. Given the presence of high heterogeneity (Table 1) and spread of study quality scores, one can conclude that study heterogeneity may be a significant factor.

## Discussion

The findings of this systematic review and meta-analysis add to the growing body of evidence supporting the hypothesis that many patient characteristics, comorbidities, symptoms, and vital signs parameters relate to increased risk of a severe outcome or death due to COVID-19.

Presented results align well with recent systematic reviews investigating risk factors in COVID-19, highlighting that age, sex, obesity, and multiple comorbidities increase the risk of adverse outcomes [38, 66, 103–105]. This study, however, goes further than previously available literature through our mapping of a wider variety of risk variables, including symptoms and vital signs.

**Table 2.  Pooled risk estimates.**

| | Multivariate | | Univariate | |
|---|---|---|---|---|
| | *Pooled Weighted OR (95% CI)* | | *Pooled Weighted OR (95% CI)* | |
| | **Severe** | **Mortality** | **Severe** | **Mortality** |
| **Patient characteristics** | | | | |
| Male | 1.17 (0.17, 2.17) *11 | 1.87 (0.7, 3.04) *4 | 1.62 (1.29, 1.94) *17 | 1.94 (1.51, 2.37) *16 |
| Age (years) | | | | |
| >60 | 1.69 (1.25, 2.13) *10 | 3.15 (0.94, 5.36) *7 | 3.75 (2.58, 4.92) *4 | 3.04 (1.96, 4.12) *5 |
| >65 | 1.76 (1.32, 2.2) *10 | 3.79 (1.33, 6.25) *8 | 2.14 (1, 3.29) *3 | 1.89 (0.09, 3.69) *2 |
| >75 | 1.93 (1.32, 2.54) *10 | 5.82 (1.86, 9.79) *8 | - | 2.41 *1 |
| BMI | | | | |
| Obesity | 1.69 (1.13, 2.24) *7 | 1.45 (0.31, 2.59) *2 | 2.02 (1.02, 3.01) *3 | - |
| Severe Obesity | 2.07 (1, 3.13) *4 | 1.51 *1 | 1.80 *1 | - |
| Smoking | | | | |
| Active | 1.01 (0.94, 1.07) *2 | 1.21 *1 | 1.22 (0.87, 1.57) *4 | 2.13 (2.08, 2.18) *2 |
| Former | 1.31 (1.22, 1.4) *2 | - | 1.26 (1.23, 1.28) *2 | 0.56 *1 |
| History | 1.42 (1.41, 1.43) *2 | 0.83 *1 | 0.79 (0.74, 0.85) *2 | 2.06 (1.53, 2.59) *4 |
| Blood group | | | | |
| O | 0.68 *1 | - | 1.14 *1 | - |
| A | 1.45 *1 | - | 1.32 *1 | - |
| **Comorbidities** | | | | |
| Any condition | 17.48 (0.18, 34.79) *2 | - | 2.92 (1.88, 3.95) *7 | 3.24 (1.98, 4.5) *7 |
| Hypertension | 1.03 (0.86, 1.21) *9 | 1.09 (1.01, 1.16) *3 | 3.73 (2.34, 5.11) *18 | 2.44 (1.76, 3.13) *15 |
| Cardiovascular Disease | 1.09 (1.09, 1.09) *2 | 1.53 (1.24, 1.82) *4 | 3.37 (2.89, 3.85) *6 | 4.04 (1.95, 6.13) *7 |
| Chronic arterial disease | 0.94 (0.88, 1) *2 | 2.14 *1 | 2.71 (1.44, 3.98) *4 | 2.85 *1 |
| Heart Failure | 1.93 *1 | 1.43 *1 | 2.23 (1.21, 3.24) *3 | 1.91 (1.63, 2.2) *4 |
| Chronic Heart Disease | 1.52 *1 | - | 2.24 (1.68, 2.8) *4 | 5.75 *1 |
| Chronic Lung Disease | 1.52 (1.51, 1.53) *2 | 1.39 *1 | 3.54 (1.52, 5.55) *2 | 5.35 (3.81, 6.88) *4 |
| Asthma | 0.75 (0.65, 0.85) *2 | - | 0.97 (0.86, 1.08) *3 | 0.85 (0.68, 1.02) *2 |
| COPD | 1.01 *1 | 2.05 *1 | 2.47 (1.44, 3.51) *7 | 2.68 (1.8, 3.55) *7 |
| Active Cancer | 1.48 (1.26, 1.69) *3 | 2.15 (2.15, 2.16) *2 | 3.19 (2.05, 4.34) *8 | 2.4 (1.97, 2.84) *6 |
| Immunosuppression | 1.2 *1 | - | 1.17 (0.96, 1.38) *2 | 2.31 (1.96, 2.65) *2 |
| Chronic Kidney disease | 1.39 (1.13, 1.65) *3 | 1.15 *1 | 3.5 (1.4, 5.59) *7 | 2.79 (1.19, 4.4) *7 |
| **Symptoms** | | | | |

(*Continued*)

**Table 2.** (Continued)

| | Multivariate | | Univariate | |
| | Pooled Weighted OR (95% CI) | | Pooled Weighted OR (95% CI) | |
| | Severe | Mortality | Severe | Mortality |
|---|---|---|---|---|
| Fever | 1.06 *1 | 0.69 *1 | 1.98 (1.05, 2.91) *14 | 0.83 (0.69, 0.97) *12 |
| Fatigue | - | 0.86 *1 | 1.74 (1.26, 2.21) *12 | 1.33 (1.04, 1.63) *12 |
| Myalgia | 4.82 (4.63, 5.01) *2 | - | 0.82 (0.64, 0.99) *9 | 1.17 (0.93, 1.4) *6 |
| Cough | - | - | 1.58 (0.92, 2.24) *16 | 0.90 (0.73, 1.08) *13 |
| Sputum production | 11.40 *1 | - | 1.19 (0.9, 1.48) *7 | 1.33 (0.95, 1.7) *9 |
| Dyspnoea | 8.68 (8.25, 9.11) *2 | - | 7.32 (1.06, 13.57) *15 | 3.21 (2.04, 4.37) *10 |
| Chest pain | - | - | 2.41 (1.93, 2.89) *6 | 2.23 *1 |
| Nausea | 15.55 *1 | - | 1.37 (0.68, 2.06) *7 | 0.72 (0.55, 0.89) *6 |
| Diarrhoea | - | - | 1.2 (0.95, 1.46) *12 | 0.89 (0.78, 1.01) *6 |
| Pharyngalgia | - | - | 1.25 (0.86, 1.65) *6 | 0.7 *1 |
| Headache | - | - | 0.96 (0.66, 1.26) *9 | 0.95 (0.28, 1.62) *3 |
| Dizziness | - | - | 6.15 (5.36, 6.93) *3 | 1.32 *1 |
| GI Symptoms | - | - | 3.36 *1 | 1.38 (0.78, 1.99) *2 |
| Chills | 6.32 *1 | - | 1.01 (0.69, 1.33) *3 | 2.08 *1 |
| Loss of smell/taste | - | - | 1.71 *1 | - |
| Rhinorrhoea | - | - | 1.15 (0.72, 1.59) *2 | - |
| Anorexia | - | - | 3.13 (2.68, 3.57) *7 | 1.13 (0.9, 1.36) *4 |
| **Vitals** | | | | |
| Respiratory rate (≥ 24 breaths/min) | - | - | 11.6 *1 | 4.5 (2.92, 6.07) *3 |

Pooled risk estimates for patient characteristics, comorbidities, and symptoms with adverse outcomes of patients with COVID-19. * Represents the number of studies included in Pooled Weighted OR.

Prior reported literature has made it clear that certain individuals are at higher risk than others. Hence, there has been a concerted effort to profile these high-risk individuals which has resulted in the development of a variety of diagnostic and prognostic models for COVID-19, with many reporting moderate to excellent discrimination [41, 90]. Interpretation of early models, however, should be treated with caution as a result of the high risk of bias due to over-fitting, lack of external validation, low representativeness of targeted populations, and subjective/proxy outcomes in criteria for hospitalisation and treatment [105, 106]. These performance estimates may be misleading and, potentially, even harmful [105]. Efforts for future development of risk profiling should follow standardised approaches such as the

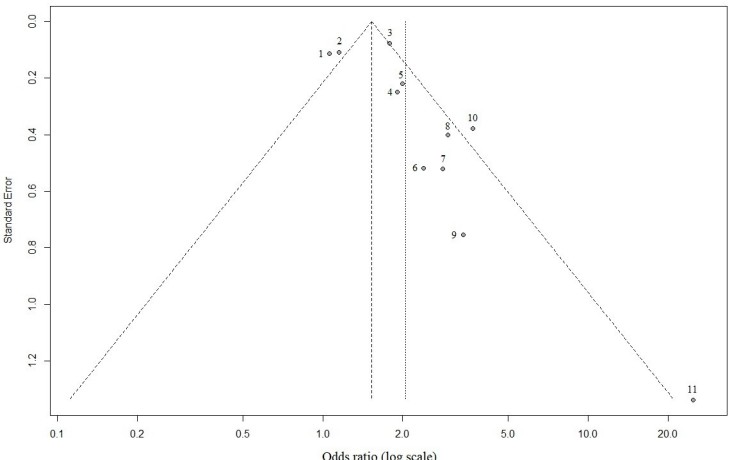

**Fig 4. Funnel plot highlighting publication bias for male sex as risk factor for severe COVID-19 outcome.**
1 = Petrilli et al., 2020; 2 = Lassale et al., 2020; 3 = Price-Haywood et al., 2020; 4 = Huang et al., 2020b; 5 = Suleyman et al., 2020; 6 = Kalligeros et al., 2020; 7 = Simonnet et al., 2020; 8 = Palaiodimos et al., 2020; 9 = Chen et al. 2020; 10 = Shi et al., 2020; 11 = Zhang et al, 2020.

TRIPOD (Transparent reporting of a multivariable prediction model for individual prognosis or diagnosis) reporting guideline [107].

The identified risk factors align with current understanding of clinical pathophysiology for severe COVID-19. There are several theories as to why age is a significant risk factor for severe COVID-19. These include the role of comorbidities, as well as decreased efficiency of the immune system related to normal ageing [108]. Male sex as a risk factor for severe disease is thought to result from a combination of the effect of health behaviours, sex hormone-mediated immune responses, and differential expression of ACE2 between sexes [109]. Obesity is a risk factor for development of comorbidities such as hypertension, cardiovascular disease, and diabetes. However, there may be further involvement of obesity through metabolic consequences, which include increased circulating cytokine levels [110].

One study included in this review stands out due to its scale, investigating the primary care records of over 17 million UK citizens [89]. Using a database of overwhelmingly unexposed individuals, the study can be differentiated from ours in that the risk associated with each variable confounds propensity for infection with the relative likelihood of death once infected. The resulting net risk weighting makes it unclear which of these two discrete probabilities is being affected by each variable. The limitation of this approach can be seen best with smoking status whereby the combined approach outputs a protective weighting, potentially due to the reported reduced infection risk conferred by active smoking, contrasting with our analysis which suggests increased prognostic risk (0.91 vs 1.21) [111]. Moreover, as the increased mortality risk of comorbidities was public knowledge before the first wave in the UK, it could be assumed that this demographic behaved more cautiously, resulting in the risk weightings being underestimated in the combined approach. Weightings for hypertension (HR: 0.88, 95% CI: 0.84–0.92 vs OR: 1.09, 95% CI: 0.86–1.37) and non-haematological cancer (using OpenSAFELY's highest risk group; diagnosed <1-year ago (HR: 1.68, 95% CI 1.46–1.94) vs our any-timeframe (OR: 2.15, 95% CI: 1.41–3.28) seem to conform to this expectation. Both approaches, however, are uniquely useful in their application and, nevertheless, are largely in alignment in their outputs. Combining the discrete risks presents the foundation for the

development of a risk model which can aid with the strategic planning required for health systems and the allocation of their resources. Our approach presents the foundation for a prognostic model which could support healthcare triage and be used on an individual level for comprehension of personal risk should one get infected.

## Limitations

While our study presents pooled findings across 14 geographies and may be considered broadly representative of the pandemic, a number of limitations should be highlighted. The primary limitation is the high heterogeneity of the included studies. Notation of patients' highest level of care may be complex to interpret because such an endpoint is dependent on local policy and resources, which have been evolving in strategy and capacity since the onset of the pandemic. Thus, a recommendation of our study is for the development of standardised protocols for reporting of COVID-19 case series and retrospective analysis. Definition of the non-severe or comparator group is often poorly defined and is likely to result in sample selection bias towards more severe cases. Recent evidence from nationwide blanket testing suggests that 86.1% of individuals who tested positive for COVID-19 had none of the three main indicative symptoms of the illness, such as cough, fever, or a loss of taste or smell [112]. In the majority of papers presented within this analysis, the individuals were already admitted to hospital, hence there is a strong selection bias towards those more severely affected and, as such, our results may underestimate the degree of risk. To facilitate rapid and widespread implementation of risk stratification, this investigation focused on risk factors that were easily obtainable. As such, we did not consider haematological risk factors within our review. These factors are known to be significant and may be valuable to include as part of risk stratification upon admission to hospital [113].

Confounding factors are highly likely in reported literature and, therefore, multivariate analysis is essential to determine causal risk factors. One such example of this is ethnicity. In our analysis of results, we chose to exclude estimates for risk relating to ethnicity and race due to the complex association of socio-economic factors and comorbidities which may be entangled with ethnicity. In early reports from the UK, there was significant disparity in outcomes for BAME (Black, Asian, and Minority Ethnic) communities [114]. However, in more recent analysis, it was found that the great majority of the increased risk of infection and death from COVID-19 among people from ethnic minorities can be explained by factors such as occupation, postcode, living situation, and pre-existing health conditions [115].

A further limitation of our study is the method used to pool risk estimates. We aimed to maximise the data collected by pulling all available estimates for risk of an associated variable. This method is flawed in that these outcomes are not directly comparable in a rigorous meta-analysis. Thus, caution is advised in interpretation of absolute risk for each variable of interest.

## Implications for future practice

A key finding of the global analysis is the difficulty in combining data reported in the literature. Healthcare systems and researchers are, at present, not providing standardised recording and reporting of health data and outcomes. This heterogeneity in reporting limits the efficacy and impact of broad meta-analysis, as highlighted by the spread of data (Fig 4). The use of standard case report forms, such as those outlined by the WHO may support this endeavour [116]. At a global level, if such data, anonymised and aggregated at patient level, is made more widely available, this could support the development of robust data-driven risk prediction models [117, 118].

At regional and provider levels, evidence-based risk stratification could help plan resources and identify trends that predict areas with increased demand. Hospital admission of severe COVID-19 cases can be expected up to two weeks following onset of symptoms [19, 119]. Hence, if risk stratification can be carried out in real-time and incorporate dynamic factors, including symptoms and vital signs, resources such as increased ICU capacity can be allocated strategically. Furthermore, through implementation of remote patient monitoring, patients can remain at home on a 'virtual ward' while under clinical observation. Early signs of clinical deterioration can be managed and, as a result, reduce hospital burden [120].

At the patient level, based on the findings of this study, it is recommended that individuals undergo comprehensive screening for risk factors including patient characteristics, detailed comorbidities, and reporting of real-time symptoms and vital sign measurements as part of a COVID-19 risk assessment. While some of the variables identified in this review are well-known risk factors within the clinical or research domain, it is essential that this information is disseminated to the general public in an easily consumable format with supporting evidence and information. The pandemic has brought about significant social and economic disruption. Due to the lack of a prior evidence-base, current guidelines for individual risk management are blunt, broad generalisations. These may be sensitive to the majority of at-risk individuals, but simultaneously have low specificity, erroneously profiling large sections of the population. Thus, the concern is that many may lose confidence in these measures, including those correctly labelled 'at-risk'. Providing individual patients with a comprehensive and individualised risk profile may empower individuals and increase engagement with public health messaging. This may facilitate efforts by national governments to encourage behaviour modification at a population level, in a manner which reduces the spread of the virus, thereby limiting socio-economic impact.

## Conclusion

The findings of this paper highlight the range of factors associated with adverse outcomes in COVID-19, across severe disease, ICU admission, IMV, and death. The determination of critical risk factors may support risk stratification of individuals at multiple levels, from government policy, to clinical profiling at hospital admission, to individual behaviour change. This would enable both a more streamlined allocation of resources and provision of support to individuals who require them most. Future studies aimed at developing and validating robust prognostic models should look to follow a standardised approach to allow for comparability and sharing of knowledge. In this respect, a continuation of open data sharing is essential to facilitate improvement of these models.

## Supporting information

**S1 Checklist. PRISMA 2009 checklist.**
(DOCX)

**S1 Fig. Forest plot for the association of patient characteristics between age and sex, and mortality in COVID-19 using a random-effects model.**
(TIF)

**S2 Fig. Forest plot for the association of comorbidities, between diabetes, hypertension and active cancer, and mortality in COVID-19 using a random-effects model.**
(TIF)

**S3 Fig. Reported risk estimates for male sex.** Size of the circle indicates sample size represented.
(TIF)

**S4 Fig. Reported risk estimates for any diabetes.** Size of the circle indicates sample size represented.
(TIF)

**S5 Fig. Reported risk estimates for any hypertension.** Size of the circle indicates sample size represented.
(TIF)

**S1 Table. Systematic literature review search terms and strategy.**
(DOCX)

**S1 File.**
(DOCX)

**S2 File.**
(CSV)

## Author Contributions

**Conceptualization:** Adam Booth, Angus Bruno Reed, Mert Aral, David Plans, Alain Labrique, Diwakar Mohan.

**Data curation:** Adam Booth, Angus Bruno Reed, Sonia Ponzo, David Plans, Diwakar Mohan.

**Formal analysis:** Adam Booth, Angus Bruno Reed, Sonia Ponzo, David Plans, Diwakar Mohan.

**Investigation:** Adam Booth, Angus Bruno Reed, Arrash Yassaee.

**Methodology:** Adam Booth, Angus Bruno Reed, Arrash Yassaee, Mert Aral, Alain Labrique, Diwakar Mohan.

**Project administration:** Adam Booth, Angus Bruno Reed, Mert Aral.

**Visualization:** Adam Booth, Angus Bruno Reed, Sonia Ponzo, Diwakar Mohan.

**Writing – original draft:** Adam Booth, Angus Bruno Reed, Sonia Ponzo, Arrash Yassaee, Mert Aral, David Plans, Alain Labrique, Diwakar Mohan.

**Writing – review & editing:** Adam Booth, Angus Bruno Reed, Sonia Ponzo, Arrash Yassaee, Mert Aral, David Plans, Alain Labrique, Diwakar Mohan.

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
