## [Decision Letter · Decision Letter 0]

19 Jan 2021

PONE-D-20-39637

Population risk factors for severe disease and mortality in COVID-19: A global systematic review and meta-analysis

PLOS ONE

Dear Dr. Plans,

Thank you for submitting your manuscript to PLOS ONE. After careful consideration, we feel that it has merit but does not fully meet PLOS ONE’s publication criteria as it currently stands. Therefore, we invite you to submit a revised version of the manuscript that addresses the points raised during the review process.

We look forward to receiving your revised manuscript.

Kind regards,

Giordano Madeddu

Academic Editor

PLOS ONE

"This research was funded by Huma Therapeutics Ltd. "

"A.B, A.B.R., S.P., D.P., A.Y., M.A., are employees of Huma Therapeutics Ltd. D.M & AL declare that they have no conflict of interests to report. "

Reviewers' comments:

Reviewer's Responses to Questions

**Comments to the Author**

1. Is the manuscript technically sound, and do the data support the conclusions?

Reviewer #1: Partly

Reviewer #2: Yes

2. Has the statistical analysis been performed appropriately and rigorously? 

Reviewer #1: No

Reviewer #2: Yes

3. Have the authors made all data underlying the findings in their manuscript fully available?

Reviewer #1: Yes

Reviewer #2: Yes

4. Is the manuscript presented in an intelligible fashion and written in standard English?

Reviewer #1: Yes

Reviewer #2: Yes

5. Review Comments to the Author

Reviewer #1: This is an interesting paper that aims at synthesizing the published evidence on relation between patients' characteristics and COVID-19 outcomes, since the beginning of pandemic until July 2020.

The focus of my review is on the methods that need some clarifications.

1. There are inconsistencies between the main text and the Supplemental material. Search strategy include PubMed, EMBASE and Web of Science in the main text, and PubMed and Scopus in Supplemental Table 1.

In Supplemental Table 1, PICO contains a description of the search terms, whereas it should report criteria for inclusion and exclusion, type of study design and choice of extracted data. The search texts for EMBASE and Web of Science are missing.

Finally, if I use [Supplemental Concept] for COVID-19 in PubMed, I just retrieve 18 articles, without using any limitation. You may have intended “MeSH”?

Using this description, I should be able to retrieve the same 2122 articles the authors found in their search, so some issue has occurred in this part of the report.

2. Publication bias. Please comment on publication bias: the funnel plot seems to indicate the presence of publication bias, at least visually. If a formal analysis has been performed, please report.

3. Through the text, odds ratios (OR) are followed by two numbers in brackets. Please state that they represent the confidence interval (95% CI I presume). In Table 2, crude and adjusted estimates should be accompanied by their CIs, so that the reader may evaluate the statistical significance of the ORs.

It would also be useful to add the number of papers reporting information included in the pooled estimates.

4. In that regard, the studies included in the Forest Plots seem far less than 76. There are seemingly discrepancies between number of studies included in Forest Plots and those with results reported in the Supplemental Tables. Please explain if a selection has been performed, and the used criteria.

Minor points:

-In the first sentence of Discussion, “…a large number…relate…” should be “relates”.

-please define CVD, RT-PCR and SD in the text

Reviewer #2: The authors conducted a review which aims to describe the relationship patient’s risk factors for severe disease and mortality in COVID-19. The research is well conducted, and it represents one of the actual hot topics in field. This is the reason why I think results should be shared with the scientific community.

There are just some minor considerations to do before publication.

Introduction

Please, provide some more information about the COVID-19 clinical features (f.e. major symptoms are fever, cough, dyspnea, and minor symptoms alter the smell and taste gastrointestinal symptoms, headache, and cutaneous manifestations. Please, read and use the following articles: https://doi.org/10.1002/hed.26269, https://doi.org/10.1002/hed.26204, https://doi.org/10.26355/eurrev_202007_22291, https://doi.org/10.1016/S1473-3099(20)30402-3, https://doi.org/10.1016/S0140-6736(20)30183-5, https://doi.org/10.1111/jdv.16669)

Methods

Please, revise the search strategy you reported. Main text and Supplemental table 1 are different in search strategy description.

When reporting Odds Ratios, please specify you are reporting specified confidence limits between the brackets.

Results

In table 2, when reporting Odds, please also provide CI limits.

Language and form

There are some typos thorough the text. Please, revise carefully before resubmission.

Before using abbreviations, please provide the definition when you use them for the first time.

After these minor revisions, the article will be ready to be published.

6. PLOS authors have the option to publish the peer review history of their article (what does this mean?). If published, this will include your full peer review and any attached files.

Reviewer #1: No

Reviewer #2: No

---

## [Author Response · Author response to Decision Letter 0]

3 Feb 2021

Please refer to the "Response to reviewers" document (we included tables and figures where necessary to better answer reviewers' queries).

---

## [Decision Letter · Decision Letter 1]

8 Feb 2021

Population risk factors for severe disease and mortality in COVID-19: A global systematic review and meta-analysis

PONE-D-20-39637R1

Dear Dr. Plans,

We’re pleased to inform you that your manuscript has been judged scientifically suitable for publication and will be formally accepted for publication once it meets all outstanding technical requirements.

Kind regards,

Giordano Madeddu

Academic Editor

PLOS ONE

Additional Editor Comments (optional):

Reviewers' comments:

Reviewer's Responses to Questions

**Comments to the Author**

1. If the authors have adequately addressed your comments raised in a previous round of review and you feel that this manuscript is now acceptable for publication, you may indicate that here to bypass the “Comments to the Author” section, enter your conflict of interest statement in the “Confidential to Editor” section, and submit your "Accept" recommendation.

Reviewer #1: All comments have been addressed

Reviewer #2: All comments have been addressed

2. Is the manuscript technically sound, and do the data support the conclusions?

Reviewer #1: Yes

Reviewer #2: Yes

3. Has the statistical analysis been performed appropriately and rigorously? 

Reviewer #1: Yes

Reviewer #2: Yes

4. Have the authors made all data underlying the findings in their manuscript fully available?

Reviewer #1: Yes

Reviewer #2: Yes

5. Is the manuscript presented in an intelligible fashion and written in standard English?

Reviewer #1: Yes

Reviewer #2: Yes

6. Review Comments to the Author

Reviewer #1: (No Response)

Reviewer #2: Dear authors, thanks for catching up my comments and for your manuscript revision.

I think the paper is now ready to be published in PLOS ONE.

7. PLOS authors have the option to publish the peer review history of their article (what does this mean?). If published, this will include your full peer review and any attached files.

Reviewer #1: No

Reviewer #2: No

---

## [Editor Report · Acceptance letter]

25 Feb 2021

PONE-D-20-39637R1 

Population risk factors for severe disease and mortality in COVID-19: A global systematic review and meta-analysis 

Dear Dr. Plans:

I'm pleased to inform you that your manuscript has been deemed suitable for publication in PLOS ONE. Congratulations! Your manuscript is now with our production department. 

Kind regards, 

on behalf of

Dr. Giordano Madeddu 

Academic Editor

PLOS ONE